# Personal Traits of the People Who Help: The Case of Bystanders to Violence against Women

**DOI:** 10.3390/ijerph192013544

**Published:** 2022-10-19

**Authors:** Andrés Sánchez-Prada, Carmen Delgado-Alvarez, Esperanza Bosch-Fiol, Virginia Ferreiro-Basurto, Victoria A. Ferrer-Perez

**Affiliations:** 1Facultad de Psicología, Universidad Pontificia de Salamanca, C/Compañía and 1-5, 37002 Salamanca, Spain; 2Facultad de Psicología, Universidad de las Islas Baleares, Ctra. Valldemossa, km. 7′5, 07122 Palma de Mallorca, Spain

**Keywords:** violence against women, gender-based violence, violence prevention, intervention

## Abstract

Within the context of emergency situations, the terms witness or bystander are used to refer to individuals involved in oppressive incidents who are neither the victim nor the perpetrator. Among the different types of emergency situations, our study focuses on violence against women (VAW). In keeping with current efforts in the scientific literature on bystander intervention and the evidence currently available, the main focus of this study is to analyze some personal factors that reflect the characteristics or experiences of bystanders and that could have a bearing on their predisposition to help victims of VAW (i.e., empathy, a just world belief system, and expectations of self-efficacy) and later analyze the possible relationship between these personal characteristics and gender or previous experience as a VAW bystander. An opportunity sample of 546 Spanish participants (73.4% women and 26.6% men) between 18 and 56 years of age took part in this study and fill out a sociodemographic data sheet, a questionnaire to evaluate the experience as violence witness designed ad hoc, and the Characteristics of People who Help Questionnaire scale (CPHQ). The results obtained indicate that CPHQ could constitute an adequate measure for the three dimensions analyzed. Female participants are significantly more empathetic than males, but in the case of a just world belief and expectations of self-efficacy the results showed no gender-related differences. Additionally, only a just world belief was clearly influenced by having been a bystander to some form of VAW. In conclusion, this study contributes a proposal for an evaluating instrument featuring three relevant personal characteristics in the development of helping behaviors, presenting some results of interest regarding empathy, a just world belief, and expectations of self-efficacy and their relationship with gender or previous experiences as VAW bystanders. These results obtained suggest an initial path toward future research in the development of interventions with bystander participation in our environment.

## 1. Introduction

Within the context of emergency situations, the terms witness or bystander are used to refer to individuals involved in an act of sexual harassment or violence but are neither the victim nor the perpetrator [1]. Particularly, in violence cases, bystanders are any individuals who observe the violence or the conditions perpetuated by such actions, who are not directly involved but are present when such actions occur, thus potentially finding themselves in a position of intervention, by providing assistance, providing an account of the events, perpetuating negative behavior, or choosing to do nothing at all [2,3]. In this regard, in the area of crime and violence prevention, as well as in much psychological research, a distinction is made between a “passive” bystander who observes a situation but neither intervenes nor takes any action (does not initiate any helpful behavior), and an “active” and/or “pro-social” bystander who intervenes or takes action in response to the observed action (initiates some type of helpful behavior) [4]. The active bystander is also called an “actionist” [5].

Among the different types of emergency situations, our study focuses on violence against women (VAW) for two main reasons:

First of all, VAW and, specifically, physical and sexual violence perpetrated by a partner or any other man constitutes a serious social and health problem of immense proportions [6,7], which leads to the pressing need to develop efficient and effective intervention strategies in these cases and to prevent future recurrence of this type of violence and its consequences [8].

Moreover, available evidence indicates the relevance of bystander responses to VAW and that social participation and helpful behavior constitute key strategies for the prevention and eradication of VAW [2,9,10,11,12,13,14,15,16]. In fact, the Council of Europe Convention on preventing and combating violence against women and domestic violence [17], opening for signature in Istanbul in 2011 and ratified by Spain in 2014, indicates (Article 27, Reporting) the importance of the role of bystanders as follows:


*Parties shall take the necessary measures to encourage any person witness to the commission of acts of violence covered by the scope of this Convention or who has reasonable grounds to believe that such an act may be committed or that further acts of violence are to be expected, to report this to the competent organizations or authorities.*


Nevertheless, and in spite of this, preventive and interventive measures based on the bystander participation are not yet widespread in Spain [14,16], which has led us to initiate a line of work in this area.

Specifically, and in keeping with current efforts in the scientific literature on bystander intervention [12], this study focuses on the review of factors related to the helpful behavior of bystanders in cases of VAW.

## 2. Personal Factors That Predispose (or Not) to Help

Although interest in prosocial behavior first began around the time of the Second World War and the Holocaust [4], the first model put forth to explain when we help is the decision-making model on bystander intervention in emergency situations [18,19], an innovative cognitive proposal to study the prosocial behavior of helping in sudden emergency situations that are impossible to foresee and require immediate action. This model proposes that the likelihood of a person helping or not helping when they encounter such a situation is dependent on a series of cognitive decisions that can be affected by various personal and situational factors, and include a series of stages, each of which leads to the decision of whether or not to intervene during the situation or at another point [20,21]: noticing the emergency; interpreting the situation as an emergency; accepting responsibility for intervening; knowing how to help; and deciding to take action.

Although many studies using this model as a referent have focused on situational factors (such as the possible ambiguity of the emergency situation or, in particular, the so-called spectator effect and the ensuing diffusion of responsibility), the fact is that, as previously noted, there are also other factors that influence the decision of whether to help (or not), including [20]: varying degrees of willingness to engage in prosocial behavior, the interpersonal attraction aroused by the victim, the prosocial models with which the spectator is familiar, the amount of responsibility the spectator attributes to the victim, etc. The present study will focus on analyzing three of the personal factors taken from the model proposed by Latané and Darley [19]. These personal factors may be in turn associated with some different variables. Of these, and considering the previous research results seen in [2] or [12], we selected two factors (gender and previous experience as a bystander) as a main focus of this study. Specifically, the reasons for this selection were that “gender is another individual-level variable that seems key for understanding helping. The helping literature more broadly finds sex differences in when and how people choose to help others” ([2] p. 218); and that “studies show that arousal in the face of distress and bystander intervention are related to an individual’s perceived similarity or connection to the victim (such us previous experiences)” ([2] p. 218).

### 2.1. Empathy

Empathy has been defined as a learned affective and cognitively complex process of understanding the emotional state or the emotional suffering of another person and includes the ability to share the emotional state of the other person, identify with that person, and understand and adopt that person’s point of view [20]. Oswald [22] clarified this definition by taking into account that empathy requires us to be capable of sharing the perspective of the other person in a dual sense: assume the cognitive perspective, that is, to know or understand what the other person is thinking and feeling, or to be able to put oneself in that person’s place (empathy in its strictest sense), and to assume the affective perspective, that is, to experience what the other person is feeling (sympathy or empathic concern). Likewise, Batson et al. [23]) established a similar distinction between a cognitive and affective perspective, although in the case of affective perspective they add a further distinction between empathy understood as experiencing how the other person feels (parallel empathy) and empathy understood as an emotional reaction in the face of the other person’s experiences and/or experiencing how one might feel themselves in that situation (reactive empathy). Several classic studies (i.e., [24]) demonstrated that empathy is one of the key personal factors to influence helping behavior, as a predictor and/or modulator of its occurrence [25,26] by modifying the motivation to act and the sense of responsibility to do so [27]. In fact, as noted by Correa [28], for some authors, empathy is a mediating behavior in any type of proactive conduct, and, as noted by Auné et al. [29], recent theories go even further by considering empathy to be one of the categories for this type of conduct and understanding that empathic motivation or predisposition among adults are not simply a correlation of a tendency toward prosocial behavior but rather are an integral part of the tendency itself (i.e., [30,31]).

Depending on the type of empathy, this could lead to different types of motivation to help. Thus, actively imagining how the other person feels and feeling their emotions would lead to empathetic interest or empathetic emotional activation, and thus to a helping behavior that is altruistic in nature, focused on alleviating or reducing the social discomfort, as argued by Batson et al. [23] in their hypothesis on altruistic empathy. In contrast, actively imagining how oneself might feel would produce empathy but also anguish, leading to a helping behavior with a self-serving motivation focused on alleviating one’s own discomfort, as argued by Cialdini et al. [32] in their negative-state relief model.

On the other hand, and in agreement with classic studies on the topic (i.e., [24]), empathy is conditioned by factors such as the characteristics of the situation; certain features attributable to the spectator, such as gender or the need for approval that enables one person to help another; or certain qualities attributable to the victim, such as likeability, attraction, or any similarity with the bystander that might induce helpful actions; etc. Although some studies on the topic corroborate these influences, it should be noted that evidence in this regard is not yet entirely conclusive [20,33].

Regarding the question of gender, some studies, as summarized by Correa [28], suggest that females are more empathetic than males, and that this could be attributed to the processes of socialization experienced in which women present a more advanced emotional development, a greater tendency toward affiliation (in contrast to a greater tendency among males toward dominance), and greater social orientation (in contrast to greater antisocial tendencies among males). Nevertheless, these differences appear to manifest to a greater degree during infancy and tend to disappear, or even reverse, with age and educational advancement [34]. Regarding the question of similarity, the affinity between the bystander and the victim in terms of personality, attitudes, experiences, etc., has demonstrated an increase in empathy and, in turn, prosocial behavior [20,21,35]. In other words, we as people tend to help to a greater degree those with whom we perceive more similarities to ourselves, given that this perception induces feelings of connection, affiliation, a link or belonging to a same group, and affection, which stimulate the desire to help. This would explain why we are more willing to help friends than strangers. However, this perception of the victim as a peer can also be counterproductive, as it may trigger feelings that the same could happen to us, which could generate negative emotions that activate defense mechanisms, such as repression (avoid or deny the threat) or awareness (concern for a threat but ability to control it, focusing on the cause of the occurrence).

It is worth noting that this similarity can also refer to the suffering experienced [35]; that is, one’s own adverse experiences may result in a greater understanding of another’s suffering, as they enhance the ability to “put oneself in another’s place”, thus increasing empathy and identification with victims among those who have suffered a similar experience, which would, in turn, influence one’s own conduct. Some studies [26] analyze “altruism born of suffering” and document the propensity of those who have been exposed to suffering (whether harm caused by other humans or by natural disasters) to help; in particular, those who have suffered similar events. Empathy is one of the mediating factors of these results. Some studies point out that this also occurs among victims of sexual abuse, who tend to exhibit increased empathy [36,37] and prosocial behavior [38] toward other victims. 

On a similar line, other studies explore the effect of being privy to a disclosure of a sexual assault and, likewise, find contradictory results. For example, Smith and Frieze [37] found that those who directly knew a victim of sexual assault had no greater empathy towards the victim than those who did not; while other studies (i.e., McMahon [39]) found that when a personal occurrence of sexual assault was disclosed to an individual, that person was better able to identify with a potential victim and this knowledge was associated with greater response rates of helping behavior. In the latter case, the ecological model of bystander intervention proposed by Banyard [2] understands that knowing a victim of sexual assault can enhance the perception of another potential victim and generate emotion and empathy for their situation, thus triggering a sense of responsibility to take action.

In general, the evidence presented indicates the existence of a strong correlation between empathy and helping behavior, suggesting the appropriateness of its inclusion as a research objective in this study.

### 2.2. Just World Belief

Any judgment and blame the bystander may attribute to the victim’s deservingness and responsibility also plays an important role when deciding whether to engage (or not) in helping behavior [21].

In this regard, it is important to bear in mind that some people maintain what is referred to as a just world belief [40]; that is, a belief system that leads them to perceive the world as a fair and equitable place where we receive what we deserve and deserve what we receive, where good deeds are rewarded and bad deeds are punished [41,42]. This constitutes a foundation for making sense of a world that is not only a fair place but also predictable [43]. Some authors (i.e., [44]) contend that in addition to a belief system relative to a just world in a general sense (even though personal experiences can be different), there is also a personal dimension relative to the belief that events that have occurred over time in our lives are fair, even though it may not have been the case for others (for an empirical example, see [45]).

Since evidence of undeserved suffering would weaken this belief system and cause discomfort, those who sustain such a belief tend to defend their position by shifting blame to the victim; that is, they tend to blame the victim for what happened to them, concluding that they were the cause of their own misfortune because if it happened to them they deserved it for “something (bad) they must have done” [41].

Available evidence seems to suggest that holding these beliefs could lead to inappropriate negative reactions and social judgments of those who are thought to be victims or find themselves in a situation of disadvantage [42], including, for example, blaming victims of sexual harassment [46], intimate partner violence [47] or sexual violence for the assault they suffered [48], or showing fewer positive attitudes towards the victims of rape [49]; all of which could, in turn, influence the predisposition of the bystander not to intervene [50,51,52]. On the other hand, other studies (i.e., [53]) did not find any relationship between these beliefs and attribution of blame. In fact, other authors provide evidence suggesting that the assumption of a just world belief (especially a personal belief) correlates to prosocial and altruistic behavior in the sense that certain individuals who adhere strongly to these beliefs are more willing to help those who need it, to perceive altruism among others, and to more highly rate the concept of interpersonal trust [54].

Many contradictory results have been obtained regarding the relationship between the idea of a just world belief and gender, with some studies not finding differences between genders regarding this construct [42,55,56,57,58], while others [47] observed that these beliefs were stronger among men, and others still [59] that it was women who demonstrated these beliefs to a greater extent.

Research on the idea of a just world belief has found evidence that social identification is an important factor for explaining the threat that innocent victims pose to certain individuals adhering to this belief system [60]. In this sense, there is evidence to suggest that defending one’s beliefs by shifting blame to the victims (blaming them for what happens to them) is especially likely when bystanders who could help do, in fact, feel personally threatened by the traumatic situation taking place [61], when they perceive the victims as their peer [62], or when the situation experienced by the victims is particularly unjust [61]. In short, there are contradictions in the relationship between the idea of a just world belief and helping behavior and the roles of various factors, such as gender or prior knowledge of VAW cases. As more extensive research in this area is necessary, this concept has been included as an objective in this study.

### 2.3. Expectations of Self-Efficacy

The penultimate stage in the decision-making model on bystander intervention in emergency situations [19] refers to knowing how to act, in other words, to an assumption that the bystander possesses the abilities needed to be able to safely intervene in a wide array of situations. However, it is insufficient to simply possess these abilities; an integral part of the decision-making process is that bystanders must also be confident in their abilities and capacity to help, that is, they must have expectations of self-efficacy [2].

There is ample evidence in this context that expectations of self-efficacy serve as strong predictors [13,29,63,64] and modulators [63] of prosocial behavior while, in contrast, perceiving a deficiency in one’s own abilities constitutes a significant barrier to help [50].

The relevance of these expectations with regard to the prevention of sexual aggression by means of helpful behavior led Banyard et al. [14,65,66] to create a scale to measure the self-efficacy of the bystander. Its use indicated [65] that the greater the confidence of the bystander in their ability to intervene in the prevention of a sexual assault, the greater their intention to develop helping behavior, and the greater the frequency with which the bystander reports to have engaged in such behavior. 

Regarding gender differences given these constructs, research thereon indicates contradictory results: while some found no differences between women and men in terms of bystander efficacy and responsibility [66]), others observed that young girls and women, as a group, have significantly higher levels in both constructs [50,67], suggesting the need for further research in this area. 

Pertaining to the question of prior experience with VAW and its relationship with self-efficacy, research on the topic [65,68] indicates that girls in general present higher levels of self-efficacy when they had previously witnessed or experienced VAW.

## 3. Current Study

Given the evidence currently available and previously presented on the predisposition to help victims of VAW [12], the main objective of this study is to analyze some personal factors that reflect the characteristics or experiences of bystanders (such as empathy, a just world belief system, or expectations of self-efficacy), which could have a greater bearing on their predisposition to help victims of VAW [12], and later examine the possible relationship between these personal factors and gender or previous experience as a bystander of this type of violence.

Obviously, there a number of instruments to evaluate these personal factors, including some validated in Spanish. Among the vast array of instruments to evaluate prosocial behavior (for a review, see [29,69]), some (i.e., Batería de Personalidad Prosocial de Penner et al. [70]; Prosocial mediation scale for adults by Caprara et al. [30]; Prosocial–antisocial behavior mediation scale in daily life and traffic-related situations by López de Cózar et al. [31]; Prosocial behavior Questionnaire by Martorell et al. [25]; Prosocialness scale for adults Auné et al. [71]) include a scale to evaluate empathy. There are also specific measures for this purpose, such as the Interpersonal Reactivity Index (IRI) [72], validated in Spanish by Mestre et al. [73]. There is even a Rape Empathy Scale by Deitz et al. [74]. Regarding the notion of a just world belief, the Global Belief in a Just World Scale (GBJWS) [75] has been adopted and used in Spanish both in Spain [47] and Latin America [76]. In the case of self-efficacy, there is, as previously noted, a specific scale to measure bystander self-efficacy [9,65,66], although it has not yet been validated in Spanish.

Among the available instruments, one questionnaire was specifically designed to measure these characteristics: the Characteristics of People who Help Questionnaire (CPHQ) [77], proposed by its authors as part of the results from previous research on traits identified as good predictors of helping behavior. Given that this instrument was viewed as an appropriate means for meeting the core objective of this study, despite the lack of information related to its psychometric characteristics, the analysis of these characteristics constituted a prior objective of this study.

## 4. Materials and Methods

### 4.1. Participants

This study analyzed a convenience sample of 546 participants between 18 and 56 years of age (M = 22,02, S.D. = 4.04) composed of 401 women (73.4%) and 145 men (26.6%), with a similar average age, although the average age was slightly higher among men (M = 22.60, S.D. = 4.46) than women (M = 21.80, S.D. = 3.86), t (541) = 2.037, *p* = 0.042. Of all the participants, three (0.5%) had completed studies in primary education, 309 (56.6%) in secondary education, 66 (12.1%) in intermediate level Vocational Training, and 168 (30.8%) at a university level.

### 4.2. Instruments

The information was gathered by means of a questionnaire that included:Questionnaire to evaluate the experience as violence witness designed ad hoc, which included a four-point answer scale (1: No, never; 2: Yes, on one occasion; 3: Yes, on more than one occasion; and 4: Yes, regularly), with the participants having been witness to 4 types of violence: robbery, intimate partner violence against women (defined in the questionnaire as physical, psychological, and/or sexual aggression against a woman perpetrated by her male partner or ex-partner), street harassment (defined in the questionnaire as unwanted comments, whistling, or touching that had taken place in the street or in public places or public transportation) and sexual harassment (defined in the questionnaire as verbal or physical sexual behaviors, such as comments, jokes, touching, etc., that had taken place at work or in an academic setting). It should be noted that intimate partner violence against women and sexual harassment are defined according to the Council of Europe Convention on preventing and combating violence against women and domestic violence [17] and Spanish laws; and street sexual harassment defined as in previous research (see Ferrer et al. [78]).Characteristics of People who Help Questionnaire scale (CPHQ) [77]. This self-report scale consisted of 20 dichotomous (true or false) items that measured predictive traits of helping behavior. The respondents were meant to indicate whether they considered characteristics of each item to define them.

### 4.3. Procedure

A non-probabilistic convenience sample was used. The questionnaire for gathering information was implemented in March–April 2021 on the Lime Survey platform and was disseminated through the social networks used by the research team and their collaborators (e.g., Twitter). A text explaining the objectives and conditions of the study was included at the beginning and access to the answer form implied preliminary acceptance on behalf of the participants to take part in the study.

### 4.4. Data Analysis

The factorial structure of the CPHQ was determined by an Exploratory Factor Analysis (EFA) on the matrix of tetrachoric correlations, given the nature of the data, using the RULS estimation method that, according to Yang-Wallentin, Jöreskog, and Luo [79], is a robust variant of the unweighted least squares (ULS) method used to estimate the parameters. Factors were identified by applying a multiple criteria analysis, which included an optimized variant from a Parallel Analysis [80,81], an analysis of the factorial commonalities and load, the conceptual item-factor coherence, and the goodness-of-fit comparative analysis of the models.

In order to evaluate the fit among the different models obtained in the EFA, the following robust indices were used [82,83,84]: for measures of absolute adjustment, the goodness of fit index (GFI), the adjusted goodness of fit index (AGFI), and the Root Mean Square Error of Approximation (RMSEA) were used. For measures of incremental adjustments, the comparative fit index (CFI) and the non-normed fit index (NNFI) were used.

The internal consistency of the CPHQ dimensions was calculated by KR20 formula. It should be noted that, given the dichotomous nature of the items, both KR20 and Cronbach’s Alpha could also be appropriate, so Cronbach’s formula is an extension of KR20 suitable for both dichotomous and polytomous data. Indeed, both formulas yield the same estimates, since the CPHQ items are scored 1/0 (see Cho and Kim [85]; Sijtsma [86]).

The Student’s *t*-test was applied to independent samples to analyze possible differences between men and women, as well as between the different experiences as violence witness, with respect to the characteristics of the persons who provided help (CPHQ). Scores were obtained for each of the CPHQ factors previously identified through a factorial analysis. In cases of non-homoscedasticity, contrasted through the Lévène *t*-test, a value of *t* with non-homogenous variances was used, once adjusted for the degrees of freedom.

A descriptive analysis was applied to the frequency of experiences as violence witness among the sampling and its relationship with regard to gender was explored by means of a chi-squared test. The relationship between the frequency of experiences as violence witness and the characteristics of those who had helped was analyzed with a Pearson correlation analysis. Finally, two multiple regression analyses using the stepwise method were used to explore whether the experiences as violence witness analyzed significantly predicted the characteristics of those who helped between men and women.

The EFA was performed using the FACTOR 10.8 program [87,88].

Subsequent analyses were performed with the SPSS 23 program (SPSS Inc., Chicago, IL, USA).

## 5. Results

### 5.1. Analysis of the Internal Structure of CPHQ

To begin, an initial EFA was conducted on the CPHQ with a Promax rotation, given the possibility of correlation factors. The previous parallel analysis suggests the existence of 4 factors that explain 48.70% of the variance prior to the rotation, with a poor adjustment of the data (CFI = 0.929; NNFI = 0.884). Of the 20 items that compose the scale, four (items 2, 4, 8, and 13) present very low communalities and a factorial load <0.30 among the factors, for which reason they were eliminated from the analysis. Given the absence of correlation observed among the factors, a new EFA with a Varimax rotation was conducted on the 16 remaining items resulting, once again, in a four-factor structure that explains, in this case, 58.83% of the variance, although this time with a good adjustment of the model. After observing low communalities among 3 items (3, 7, and 12) a new EFA was applied, eliminating the items from the analysis due to their low contribution to the factorial structure. The result was a new four-factor structure that explains, in this case, 65.54% of the variance with better adjustment indices. After observing one item (18) with a low contribution to the structure (communality <0.30), it too was eliminated. The final four-factor structure obtained explains 68.84% of the variance, with an adjustment of the model similar to the previous model in which the 12 remaining items exhibit acceptable communalities (≥0.30). Table 1 presents a comparison between the different models in the CFA.

As the final factorial solution exhibits a factor of only two items, new factorizations were applied until a factorial solution of 69.30% was finally obtained.

As can be seen in Table 2, Factor 1 is composed of three items (6, 17, and 20) relative to the just world belief with factorial loads greater than 0.65 and an internal consistency of 0.542; Factor 2 is composed of three items (9, 10, and 14) relative to empathy and care orientation, with factorial loads greater than 0.55 and an internal consistency of 0.326; and Factor 3 is composed of 3 items (5, 11, and 19) relative to the expectations of self-efficacy with loads greater than 0.70 in two of the items and one load >0.40 (item 19) and an internal consistency of 0.464. It should also be noted that, although the factorial load of item 19 was greater in Factor 1, it was considered as part of Factor 3 due to its theoretical coherence. Ultimately, and according to these results, the greatest strength derived from EFA is the theoretical clarity of the factors obtained; on the other hand, the greatest weaknesses were the limited adequacy of the sample in all the analyses and the low internal consistency of the factors obtained.

It is important to note that, despite the previously mentioned weak points, the adjustment of the model obtained in the FCA is excellent and superior to the model obtained from previous analysis (GFI = 0.988; AGFI = 0.965; CFI = 0.999; NNFI = 0.996; RMSEA = 0.019 (0.005, 0.032)).

### 5.2. Characteristics of Bystanders and Gender

Once the factors composing CPHQ were determined, the next step involved analyzing whether significant differences were present between men and women with respect to those factors.

As can be seen in Table 3, there are no statistically significant differences with regard to either a “just world belief” or “expectations of self-efficacy”. Only in the case of “empathy and care orientated” were significant differences observed (*p* < 0.001) with women obtaining significantly higher scores in this factor.

### 5.3. Characteristics of Bystanders Who Help and Experiences as Violence Witnesses

To analyze the differences among the factors comprising CPHQ among people having witnessed different types of violence, the variable was dichotomized into two categories: having been a bystander or not having been a bystander of any of the types of violence studied.

The results obtained (Table 4) show that having been a bystander to some form of violence is only related to belief in a just world. Specifically, those who had been a witness to intimate partner violence against women or sexual assault exhibit significantly less belief in a just world than those who had not suffered such an experience.

### 5.4. Differences between Men and Women in Experiences as Violence Witnesses

The next step was to apply a descriptive frequency analysis to the different experiences as violence witness studied.

As can be seen in Table 5, except in the case of theft, the frequency distribution for the experiences as violence witness studied reveals a clearly differentiated pattern between men and women. There is, therefore, a clear relationship between both variables in all forms of VAW, as women have been bystanders to this type of situation much more frequently than men.

### 5.5. Relationship between the Characteristics of People Who Help and Experiences as Violence Witnesses

Given the existence of a differentiated pattern for VAW experiences as a witness, the next step was to analyze, for women and for men separately, the relationship between the characteristics of people who help and their experiences as a witness of violence.

The results obtained (Table 6) show that there is no relationship between the experience of being a bystander to some form of common violence, such as theft, and any of the characteristics attributable to the helping people analyzed in this study. In contrast, with respect to the experience of witnessing some form of VAW, there was a negative and significant relationship among women bystanders to any of the three forms of VAW analyzed (intimate partner violence, sexual assault, and street harassment) and a just world belief and between expectations of self-efficacy and having been a bystander to street harassment. However, among men, a significant, and also negative, relationship is only present between a just world belief and having been a bystander to sexual assault.

In order to study this relationship in greater depth, a regression analysis by steps was performed (Table 7), establishing that, in the case of both women and men, having been a bystander to street harassment was the only one of the experiences as violence witness studied that served as a predictor of any of the characteristics of the helping people studied, specifically the notion of a just world belief.

## 6. Discussion

With respect to the previous objective formulated for this study, the results obtained allow us to conclude that CPHQ could constitute an adequate measure for three of the dimensions that, according to the psychosocial literature, are relevant to an understanding of the helping behavior of a bystander: belief in a just world, empathy and being care oriented, and expectations of self-efficacy. In fact, the main strength of this instrument is precisely, as is confirmed from the results, the theoretical clarity of the component factors. In contrast, its primary weakness is the low internal consistency of these factors. It should be noted that this low internal consistency could be related to the low number of items per factor. In this sense, it is necessary to point out the low number of items, taking into account that, in general, “reliability of item clusters, say three or four items, is notoriously low, at best usually around 0.30–0.40” [89] (p. 156). Another possible explanatory factor for the low consistency could be the use of a dichotomous response scale. Given that both the theoretical clarity and the low number of items per factor could serve as a time-saving advantage when looking to use it with a range of scales, for future use we propose substituting the dichotomous response scale currently used for a 5 point or 7 point Likert type scale used to express how much the people agree or disagree with the items in order to improve its internal consistency.

With respect to the main focus of this study, first of all, when looking at the factors evaluated by the CPHQ, the results obtained indicate that female participants of this study are significantly more empathetic than males, in line with the suggestions put forth by Correa [28], who indicated that these differences could be related to the processes of differential socialization and their corresponding effects. In the case of a just world belief and expectations of self-efficacy, our results are in line with those from previous studies, showing no gender-related differences in either the first (i.e., [42,55,56,57,58]) or the second [65,66] of these factors.

Secondly, with respect to prior experiences as violence witness, only a just world belief is clearly influenced by having been a bystander to some form of VAW. This influence was apparent in the different analyses performed (a comparison of medians, correlation, and regression). While the expectations of self-efficacy indicate a relationship, this occurred only for bystanders of some form of VAW (sexual assault) and only among women.

The absence of a relationship between prior experience as a bystander to VAW and empathy is in agreement with the findings of Smith and Frieze [37], who found that individuals who personally knew a victim did not demonstrate greater empathy than towards victims who they did not know. However, as summarized by Rojas-Ashe et al. [15], a diversity of studies demonstrate how receiving firsthand information about violence helps people to identify with a potential victim and also to correctly identify a situation of violence. Given that these factors are associated to higher rates of helping behavior, further research is required to study the possible effects of receiving firsthand information or being a bystander to VAW regarding variables such as empathy, the perception of possible scenarios of such violence, and, where appropriate, engaging in helping behavior.

Regarding expectations of self-efficacy, the inverse and significant relationship observed between this factor and having been a bystander to sexual assault is contrary to the results identified in the previous literature on the subject (i.e., [65,68,90]). These contradictions suggest the need for further studies to delve into the matter and investigate the possible explanations behind these results.

Finally, with respect to a just world belief, the results obtained indicate that those who have been bystanders to intimate partner violence and sexual assault tend to exhibit this belief to a lesser degree than those who have not. Moreover, there is a significant and negative correlation among women between a just world belief and a bystander to any of the forms of VAW studied, whereas among men the correlation is between this belief and having been a bystander to street harassment. In the cases of both men and women, having been a bystander to street harassment is the only one of the experiences as violence witness in this study that served as a predictor of any of the characteristics among the subjects studied, specifically the just world belief. In the end, among the personal factors under study, a just world belief was most directly related to the experiences as violence witness endured by the participating subjects. This result suggests the need to further delve into this factor from different perspectives (e.g., analyzing their relationship with other variables, such as religiosity or social dominance).

These results, therefore, call for a more in-depth study on the attribution of responsibility in the case of victims of VAW, incorporating the concept of a moral disconnection or disassociation [91], which is framed within a cognitive social theory and can be described as the set of cognitive processes or beliefs regarding the attribution of responsibility, damage caused and the victims who allowed their aggressors to rationalize or justify their harmful or wrongful conduct. As previously noted, people holding a strong just world belief tend to adopt cognitive strategies that include the devaluation of the victim or an assumption of their guilt, allowing the bystander to minimize the injustices they witness and reduce their anxiety as spectators [92]. These cognitive strategies can be closely associated with certain moral disconnection or disassociation mechanisms, which requires a detailed examination of the role of moral disconnection mechanisms and their relationship with this type of belief, in line with the research of De Caroli and Sagone [54] and Shen et al. [93].

Additionally, as noted by Rojas-Ashe et al. [15], various studies suggest that the acceptance of myths about rape (i.e., false cultural beliefs about rape, victims of rape, and aggressors), which often serve to blame the victims and exonerate their aggressors, are related to a just world belief [48,49,94], despite being different concepts [95]. For this reason, it would also be important to look more closely into an analysis in this context, especially when previous research has linked higher levels of these myths with lower levels of feelings of responsibility, helping attitudes, and helping behavior [39,66].

It is also important to note that the relationship between a just world belief system and helping behavior not only fits into the decision-making model on bystander intervention in emergency situations, as can be seen in Latané and Darley [19], but is also key to the conceptual model on the impact of social justification for VAW as presented by Waltermaurer [52], given that this justification impacts the perpetration, experiences as a violence witness, and response to this violence in societies where VAW is perceived as justifiable, making it more likely for the abuser to act and for the victim not to file a report, and for any potential bystander not to intervene. Results relative to this factor are therefore of particular importance.

## 7. Conclusions

In conclusion, this study provides a proposal for an evaluating instrument featuring three personal characteristics relevant in the development of helping behaviors, presenting some results of interest regarding empathy, a just world belief, and expectations of self-efficacy and their relationship with gender or previous experiences as VAW bystanders. In fact, the results obtained offer an improvement of the knowledge regarding the subject in our context and suggest an initial path toward future research in the development of interventions with bystander participation in our environment.

Nevertheless, and in spite of it, this study is not without limitations. Thus, and with a reference to the sample, it is worth noting that this was an incidental sample, composed mainly of women and young adults (97.4% were less than 30 years of age). For this reason, continued research is necessary to determine the extent to which these results can be applicable to other populations, particularly different age cohorts. Moreover, using a single questionnaire (CPHQ) and, in particular, having limited, adequate data for AFE and low internal consistency of the factors obtained are also a limitation in this study, which could be compensated in future research by modifying the answer scale and comparing these results with those that apply other questionnaires designed for measuring the three factors included. Further research is needed regarding the relationship among the variables analyzed and between gender and the experiences as a bystander or witness to VAW, given that the results obtained suggest avenues for future research, as previously noted. However, in any case, these contributions may be relevant in the pursuit of developing efficient preventive and interventive practices based on bystander participation [10,11,14].

## Figures and Tables

**Table 1 ijerph-19-13544-t001:** Comparison of CPHQ models with 4 factors. Fit indices.

Criteria	Good/Acceptable	Fit AFE 2 (4 Factors/16 Items)	Fit AFE 3 (4 Factors/13 Items)	Fit AFE 4 (4 Factors/12 Items)
GFI	≥0.95/≥0.90	0.966	0.977	0.978
AGFI	≥0.90/≥0.85	0.935	0.944	0.940
CFI	≥0.97/≥0.95	0.976	0.988	0.989
NNFI	≥0.97/≥0.95	0.953	0.970	0.970
RMSEA [95% IC]	≤0.05/≤0.08	0.047 [0.018, 0.051]	0.041 [0.018, 0.037]	0.042 [0.016, 0.047]

**Table 2 ijerph-19-13544-t002:** Factorial structure of CPHQ.

	F 1	F 2	F 3	*h* ^2^
6. In general, I believe that everybody has what they want or what they deserve.	0.771			0.660
17. Sooner or later, good deeds are rewarded and bad deeds are punished.	0.655			0.441
20. Each person decides their own fate.	0.772			0.629
9. The welfare of people takes precedence over all things.		0.556		0.377
10. I feel sad when I hear about the misfortunes of others.		0.770		0.602
14. Doing something for somebody else makes me feel really good.	0.316	0.606		0.474
5. I consider myself to be a competent and efficient person.			0.854	0.737
11. I believe in myself.			0.735	0.564
19. I can achieve whatever I set my mind to.	0.535		0.417	0.469
Internal consistency (KR-20)	0.542	0.326	0.464	

Note: The table does not include factorial loads < 0.30.

**Table 3 ijerph-19-13544-t003:** Differences in characteristics of the people who help by gender.

		Men (n = 145)	Women (n = 401)	*t*
Belief in a just world	Average s.t.	1.30 (1.01)	1.18 (0.98)	*t* (544) = 1.318 *p* = 0.188
Empathy and care orientation	Average s.t.	2.72 (0.59)	2.89 (0.35)	*t* (181.99) = −3.398 *p* < 0.001
Self-efficacy expectations	Average s.t.	2.12 (0.90)	2.16 (0.90)	*t* (544) = 0.487 *p* = 0.627

**Table 4 ijerph-19-13544-t004:** Differences in characteristics of the people who help by experiences as violence witness.

	Having Witnessing		Average (s.t.)	*t* *p*
Belief in a just world	Robbery	Yes (n = 236)	1.14 (0.96)	*t* (458) = 0.031 *p* = 0.975
		No (n = 224)	1.14 (0.98)	
	Intimate partner violence	Yes (n = 335)	1.10 (0.95)	*t* (422.15) = 3.136 *p* = 0.002
		No (n = 210)	1.38 (1.02)	
	Sexual harassment	Yes (n = 356)	1.15 (1.00)	*t* (543) = 1.991 *p* = 0.047
		No (n = 189)	1.32 (0.97)	
	Street sexual harassment	Yes (n = 456)	1.18 (1.00)	*t* (543) = 1.448 *p* = 0.148
		No (n = 86)	1.35 (0.94)	
Empathy and care orientation	Robbery	Yes (n = 236)	2.85 (0.44)	*t* (458) = 0.702 *p* = 0.483
		No (n = 224)	2.88 (0.39)	
	Intimate partner violence	Yes (n = 335)	2.87 (0.39)	*t* (372.98) = −1.289 *p* = 0.198
		No (n = 210)	2.81 (0.49)	
	Sexual harassment	Yes (n = 356)	2.87 (0.40)	*t* (330.76) = −1.362 *p* = 0.174
		No (n = 189)	2.81 (0.48)	
	Street sexual harassment	Yes (n = 456)	2.86 (0.41)	*t* (105.38) = −1.561 *p* = 0.122
		No (n = 86)	2.77 (0.52)	
Self-efficacy expectations	Robbery	Yes (n = 236)	2.15 (0.88)	*t* (458) = −0.174 *p* = 0.862
		No (n = 224)	2.13 (0.89)	
	Intimate partner violence	Yes (n = 335)	2.14 (0.86)	*t* (407.89) = 0.208 *p* = 0.836
		No (n = 210)	2.16 (0.96)	
	Sexual harassment	Yes (n = 356)	2.10 (0.89)	*t* (543) = 1.734 *p* = 0.084
		No (n = 189)	2.24 (0.91)	
	Street sexual harassment	Yes (n = 456)	2.14 (0.90)	*t* (543) = 0.703 *p* = 0.482
		No (n = 86)	2.21 (0.88)	

**Table 5 ijerph-19-13544-t005:** Differences in frequency of experiences as violence witness by gender.

		Never	Once	More than 1	Habitually	χ^2^ (3 df)
Robbery n = 453	Men n = 120	50 (41.7%)	35 (29.2%)	34 (28.3%)	1 (0.8%)	7.37 *p* = 0.061
	Women n = 179	171 (51.4%)	96 (28.8%)	66 (19.8%)	-	
Intimate partner violence n = 539	Men n = 133	83 (58.9%)	20 (14.2%)	30 (27.0%)	-	38.75 *p* < 0.001
	Women n = 396	122 (30.8%)	112 (28.2%)	145 (36.6%)	17 (4.3%)	
Sexual harassment n = 537	Men n = 141	64 (45.4%)	24 (17.0%)	49 (34.8%)	4 (2.8%)	22.62 *p* < 0.001
	Women n = 396	124 (31.2%)	37 (9.3%)	196 (49.5%)	39 (9.8%)	
Street sexual harassment n = 537	Men n = 141	42 (29.8%)	18 (12.8%)	70 (49.6%)	11 (7.8%)	57.39 *p* < 0.001
	Women n = 396	40 (10.1%)	30 (7.6%)	187 (47.1%)	139 (35.0%)	

**Table 6 ijerph-19-13544-t006:** Relationship between characteristics of people who help and experiences as violence witness.

		Witnessing a Robbery	Witnessing Intimate Partner Violence	Witnessing Sexual Harassment	Witnessing Street Sexual Harassment
Women (n = 336)	Belief in a just world	−0.041	−0.095 *	−0.110 *	−0.147 **
	Empathy and care orientation	0.046	0.032	0.086	0.064
	Self-efficacy expectations	−0.001	−0.056	−0.093 *	−0.017
Men (n = 124)	Belief in a just world	0.050	−0.137	−0.142	−0.230 *
	Empathy and care orientation	−0.074	−0.084	−0.018	0.105
	Self-efficacy expectations	−0.068	−0.102	−0.138	−0.062

* *p* < 0.05; ** *p* < 0.01.

**Table 7 ijerph-19-13544-t007:** Predictors of the characteristics of people who help.

	Criteria	Predictor (*p* < 0.05)	R^2^	*ϐ*	*p*
Women (n = 336)	Belief in a just world	Witnessing street sexual harassment	0.022	−0.147	0.007
Men (n = 124)	Belief in a just world	Witnessing street sexual harassment	0.053	−0.230	0.010

## Data Availability

The raw data supporting the conclusions of this manuscript will be made available by the authors on request.

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
