# Peer review of "Personal Traits of the People Who Help: The Case of Bystanders to Violence against Women"

_ijerph, 2022, doi:10.3390/ijerph192013544_

Round 1
Reviewer 1 Report
The manuscript analyses the potential for the CPHQ scale to evaluate personal characteristics of helping behavior, with the objective to establish the main personality traits of bystanders. Furthermore, it adds other variables such as: having witnessed violence and the role of gender.
This is an interesting research that adds to the extensive research on bystanders that has been published in the later years. The manuscript is well-written and the methods and results are clearly stated. Despite this, the discussion could be improved in order to enhance the implications of this study. In what follows are my comments:
- Introduction should be shortened as the explanation of each dimension is rather long. Moreover, previous research that used the CPHQ scale and the main results obtained should be added, so that readers familiarize themselves with the tool and its potential already in the introduction section. Latest studies on bystander responses to gender-based violence should also be included.
- Materials and methods: I don't understand why it was decided to differentiate between IPV against women, street harassment, and sexual harassment. What is the rationale behind using those categories? how is IPV different from sexual harassment? Is sexual harassment understood as happening outside the couple? Were those categories previously described to the participants? If yes, this description should be added to the manuscript.
- Discussion: in the discussion I missed some further implications and discussions of the reasons and hypotheses for the results obtained. What are the implications of the relationship between having been bystanders and having lesser belief in a just world? What are the implications in relation to victim blaming and bystanders' responses? And what are the implications for the result that this correlation is different between men and women depending on the type of violence they have witnessed? The same is needed for the results on the expectations of self-efficacy. Results on empathy should be discussed taking into account the complexity of this construction, especially when your study this construct is analyzed just with three items. This should be considered in the limitation section also.
Reviewer 2 Report
My comments and suggestions for authors are detailed in the document attached.

Reviewer 3 Report
The topic of study is very interesting and useful.
If a qualitative research method was used to develop the tool (for example, directed content analysis), it could reduce the limitations of the research.
Round 2
Reviewer 2 Report
This is a revised version of a research manuscript analyzing the impact of respondents’ some personal factors (i.e., empathy, a just world belief system, and expectations of self-efficacy) in their predisposition to help victims of violence against women (VAW). I appreciate the changes made by the authors and consider that they address the core of my suggestions. Still, I have some issues about their research:
Current Study
- Regarding the respondents’ personal variables that the authors included in the analyses (i.e., sex and having had witnessed violence experiences), I was wondering whether other personal characteristics might have a greater bearing on the factors they analyze (for example, religiosity and social dominance orientation in case of just world belief system). In this regard, it is recommended that the authors better justify the criteria to select these two variables (and no others).
Materials and Methods
- Did the respondents receive the detailed description of the three types of violence against women studied? What was the exact wording of the questions included in the survey? This is important because it might affect the report rates.
- There is a typo in word “Twitter” (line 405).
General Concerns:
I understand that the authors do not analyze the respondents’ predisposition to help victims of VAW. However, they indicate that they “analyze some personal factors (i.e., empathy, a just world belief system, and expectations of self-efficacy) that reflect the characteristics or experiences of bystanders that could have a greater bearing on their predisposition to help victims of VAW”. I am afraid that they, in fact, analyze some personal factors that might influence the predisposition to help victims in general, since there is no reference to VAW in the CPHQ (at least to my knowledge, see Table 2).
We already know that the predisposition to intervene in violent situations depends on a broad list of factors. Among them, the type of offense is very relevant, and the specific context of violent situation happens within is also a key factor (in addition to others such as the relationship with the victim, the number of bystanders…). What I am trying to understand is why the authors apply their results to the specific field of VAW while they did not include any reference to this form of violence. It seems to me that they contribute to the literature in a broad sense, but not in the specific field of VAW.
Author Response
Dear Reviewer,
Thanks to you for the feedback about our paper (IJERPH-1912902R1).
According to your suggestions, we have revised and changed our paper “Personal traits of the people who help: the case of violence against women bystanders" (IJERPH-1912902R1) in order to improve it. Specifically we have made the following changes:
This is a revised version of a research manuscript analyzing the impact of respondents’ some personal factors (i.e., empathy, a just world belief system, and expectations of self-efficacy) in their predisposition to help victims of violence against women (VAW). I appreciate the changes made by the authors and consider that they address the core of my suggestions.
Author’s reply: It could be noted that our manuscript doesn’t analyze the impact of respondents’ some personal factors (i.e., empathy, a just world belief system, and expectations of self-efficacy) in their predisposition to help victims of violence against women (VAW).
In fact, the main focus of our study is to analyze some personal factors that reflect the characteristics or experiences of bystanders. According to the literature on the subject, these personal factors could have a bearing on the bystanders’ predisposition to help victims of VAW, but we don’t analyze this predisposition. And, additionally, we analyze the possible relationship between these personal characteristics and gender or previous experience as a VAW bystander.
We have revised and adjusted the abstract and also the objective description (see lines 453-462) to clarify this point.
Current Study
Regarding the respondents’ personal variables that the authors included in the analyses (i.e., sex and having had witnessed violence experiences), I was wondering whether other personal characteristics might have a greater bearing on the factors they analyze (for example, religiosity and social dominance orientation in case of just world belief system). In this regard, it is recommended that the authors better justify the criteria to select these two variables (and no others).
Author’s reply: Indeed, these variables suggested by the reviewer are very interesting, and their study has been added as part of the future lines of study in the discussion (see lines 925-926).
These personal factors may be in turn associated with some different variables. Among them, and considering the previous research results [see Banyard (2011) or Mainwaring et al., 2022)), we have select two (gender and previous experience as a bystander) as a main focus of this study. Specifically, the reasons for this selection were that "gender is another individual-level variable that seems key for understanding helping. The helping literature more broadly finds sex differences in when and how people choose to help others" (Banyard, 2011, p. 218); and that "studies show that arousal in the face of distress and bystander intervention are related to an individual’s perceived similarity or connection to the victim (such us previous experiences)" (Banyard, 2011 p. 218).
Banyard, V.L. Who will help prevent sexual violence: creating an ecological model of bystander intervention. Psychol Violence 2011, 1(3), 216–229, doi: 10.1037/a0023739
Mainwaring, C.; Gabbert, F.; Scott, A.J. A systematic review exploring variables related to bystander intervention in sexual violence contexts. Trauma Violence Abuse 2022, First published on line, doi: 10.1177/15248380221079660
We have included this explanation in the manuscript (see lines 122-134).
Materials and Methods
Did the respondents receive the detailed description of the three types of violence against women studied? What was the exact wording of the questions included in the survey? This is important because it might affect the report rates.
Author’s reply: We have included in the manuscript exactly the definition of each type of violence presented to the respondents in the questionnaire administered (see lines 579-592).
There is a typo in word “Twitter” (line 405).
Author’s reply: Corrected.
General Concerns:
I understand that the authors do not analyze the respondents’ predisposition to help victims of VAW. However, they indicate that they “analyze some personal factors (i.e., empathy, a just world belief system, and expectations of self-efficacy) that reflect the characteristics or experiences of bystanders that could have a greater bearing on their predisposition to help victims of VAW”.
Author’s reply: Effectively, our manuscript doesn’t analyze the predisposition to help victims. The aim of our study is to analyze three of the personal factors that predispose to help, taken from the decision-making model on bystander intervention in emergency situations proposed by Latané and Darley (see lines 120-122).
I am afraid that they, in fact, analyze some personal factors that might influence the predisposition to help victims in general, since there is no reference to VAW in the CPHQ (at least to my knowledge, see Table 2).
Author’s reply: Effectively, the CPHQ analyze some personal factors that might influence the predisposition to help victims in general. But, as the introduction expose (see section 2.1, 2.2. and 2.3), these factors are also relevant in cases of VAW bystanders. In fact, the main purpose of the research is deep on these factors because, according to the literature on the subject (and particularly, according to the Banyard’s ecological model), these personal factors could have a bearing on the bystanders’ predisposition to help VAW victims (see lines 122-134; and sections 2.1, 2.2, and 2.3).
We already know that the predisposition to intervene in violent situations depends on a broad list of factors. Among them, the type of offense is very relevant, and the specific context of violent situation happens within is also a key factor (in addition to others such as the relationship with the victim, the number of bystanders…). What I am trying to understand is why the authors apply their results to the specific field of VAW while they did not include any reference to this form of violence. It seems to me that they contribute to the literature in a broad sense, but not in the specific field of VAW.
Author’s reply: As the introduction expose (see section 2.1, 2.2. and 2.3) the personal factors analyzed are both relevant in general and in cases of VAW bystanders. And we include as an additional analysis the relation between these factors with gender and previous experience as VAW witness. In this sense, our study has the intention to make a contribution to the literature in a broad sense, but also in specific field of VAW bystanders as a previous analysis that could be completed in the future (as the discussion and conclusions sections pointed out). These are the next steps of the research we are working on.